# Avoidance of carnivore carcasses by vertebrate scavengers enables colonization by a diverse community of carrion insects

Carlos Muñoz-Lozano[1☯], Daniel Martín-Vega[2,3☯]*, Carlos Martínez-Carrasco[1], José A. Sánchez-Zapata[4], Zebensui Morales-Reyes[4], Moisés Gonzálvez[1], Marcos Moleón[5¤]

**1** Department of Animal Health, Regional Campus of International Excellence "Campus Mare Nostrum", University of Murcia, Murcia, Spain, **2** Department of Life Sciences, Natural History Museum, SW, London, United Kingdom, **3** Department of Life Sciences, University of Alcalá, Alcalá de Henares, Madrid, Spain, **4** Department of Applied Biology, University Miguel Hernández, Elche, Alicante, Spain, **5** Department of Conservation Biology, Doñana Biological Station (EBD-CSIC, Seville, Spain

☯ These authors contributed equally to this work.
¤ Current address: Department of Zoology, University of Granada, Granada, Spain
* daniel.martinve@uah.es

**Data Availability Statement:** All relevant data are within the manuscript and its Supporting Information files.

## Abstract

Carrion resources sustain a complex and diverse community of both vertebrate and invertebrate scavengers, either obligate or facultative. However, although carrion ecology has received increasing scientific attention in recent years, our understanding of carrion partitioning in natural conditions is severely limited as most studies are restricted either to the vertebrate or the insect scavenger communities. Moreover, carnivore carcasses have been traditionally neglected as study model. Here, we provide the first data on the partitioning between vertebrate and invertebrate scavengers of medium-sized carnivore carcasses, red fox (*Vulpes vulpes* (Linnaeus)), in two mountainous Mediterranean areas of south-eastern Spain. Carcasses were visited by several mammalian and avian scavengers, but only one carcass was partially consumed by golden eagle *Aquila chrysaetos* (Linnaeus). These results provide additional support to the carnivore carrion-avoidance hypothesis, which suggests that mammalian carnivores avoid the consumption of carnivore carcasses to prevent disease transmission risk. In turn, the absence of vertebrate scavengers at carnivore carcasses enabled a diverse and well-structured successional community of insects to colonise the carcasses. The observed richness and abundance of the most frequent families was more influenced by the decomposition time than by the study area. Overall, our study encourages further research on carrion resource partitioning in natural conditions.

## Introduction

Animal carcasses are pervasive in terrestrial ecosystems due to predator kills, other natural deaths and human-caused mortality [1–6]. Despite being largely unpredictable in space and time, these nutrient-rich resources sustain a wide variety of both vertebrate and invertebrate

**Funding:** This study was partly funded by the Spanish Ministry of Economy, Industry and Competitiveness and EU ERDF funds through the projects CGL2015-66966-C2-1-2-R and CGL2017-89905-R. D.M.-V. was supported by an EC funded Marie Curie Intra-European Fellowship (FP7-PEOPLE-2013-IEF-624575) and a research contract from the University of Alcalá (Ayudas Postdoctorales UAH), Z.M.-R. by a pre-doctoral grant (FPU12/00823), and M.M. by a research contract Ramón y Cajal from the MINECO (RYC-2015-19231).

**Competing interests:** The authors have declared that no competing interests exist.

scavengers, either obligate (i.e., specialised) or facultative (i.e., opportunistic) [1–4]. Among vertebrates, obligate scavenging is only known in Old and New World vultures, but facultative scavenging is widespread within birds, mammals and reptiles [1, 7–9]. Among invertebrates, multiple insect species are specialised in the consumption of different carcass tissues [4–10], with many other carnivore, omnivore and even phytophagous insects also feeding occasionally on carrion [10–12]. In addition, the carrion insect community includes necrophilous species that do not feed on dead tissues, but are specialised predators and parasitoids of necrophagous insects [10, 13, 14]. Thus, carrion resources are directly and indirectly exploited by a potentially very diverse and complex scavenger community leading to heterotrophic succession [15].

In recent years, the role of carrion (hereafter we will specifically focus on vertebrate carrion) in maintaining biodiversity and the ecological importance of scavenging has received increasing scientific attention [16, 17, 18]. In the case of invertebrate scavenging, the study of insects associated with carrion is also important from a medicolegal and veterinary perspective [19]. Many insect species are specifically attracted to carrion during different stages of decomposition occurring in a more or less predictable chronological sequence or succession [20–22]. Therefore, if site-specific succession patterns are known, a minimum post-mortem interval may be estimated in a forensic case on the basis of the entomofauna present on the cadaver [21]. In a similar way, some studies have also explored the potential of the changes in the bacterial communities during the decomposition process of animal carcasses as indicators for death time estimations in forensic veterinary science [23]. However, two important caveats undermine current scientific understanding of carrion consumption in natural conditions, as indicated below.

First, studies simultaneously considering scavenging by both invertebrates and vertebrates are largely non-existent, with few exceptions [6, 24–26]. In addition, insect succession studies are typically carried out while protecting the experimental carcasses with wire mesh cages or other mechanisms allowing the access of insects but preventing interference by vertebrate scavengers [20, 22, 27–30]. Vertebrate scavengers can consume medium- and large-sized carcasses completely within a few hours or days [31–32] unless carcasses are covered by snow [2] or concealed. Nonetheless, their potential impact on the carrion insect community, particularly on those species that typically act as secondary colonisers, is a largely understudied issue. In addition, vertebrate scavenging research has also generally overlooked the role of the insect communities associated with carcasses [9]. Moreover, the insect scavenging activity can affect the period during which a carcass is available to vertebrate scavengers, especially in warm climates [33, 34]. Thus, interactions between vertebrate and invertebrate scavenger communities and the resulting partitioning of carrion resources are of paramount importance in scavenging ecology [25, 26], but very far from being understood [4].

Second, carrion is frequently considered an ephemeral food resource [1, 3]. While this is true in most cases, not all carcass types persist equally, even for a given carcass size. Insect succession studies are generally performed using domestic pig carcasses as surrogate of human models [29, 30, 35], and vertebrate scavenging research has mostly focussed on herbivore carcasses, mainly ungulates [1, 9] but also rodents [36, 37], lagomorphs [38, 39] and domestic chickens [31, 40]. However, the use of carrion of carnivore species (e.g., mammalian carnivores) as a study model has been traditionally neglected, with some recent exceptions [2, 39, 41]. These studies show that carnivore carcasses last considerably longer than other similar-sized carcasses. This is because carnivores largely avoid feeding on carrion of closely related species, especially of conspecifics, a behaviour likely evolved to reduce exposure to infectious pathogens [2, 41]. This avoidance behaviour leads to wide ecological and evolutionary implications. For instance, carnivore carcasses are almost fully available to vertebrate scavengers that

are phylogenetically distant from the species to which the carcass belongs, as well as to a well-structured successional community of necrophagous and necrophilous invertebrates that are not able to colonise or complete their life cycle in more ephemeral carcasses [41]. Moreover, the insect scavenging fauna associated with wildlife vertebrate carrion, and more specifically with carnivore carcasses, remains largely unstudied, with the exception of a few studies using generally very few (one or two) carcasses [27, 28]–a typical sample size in many forensic studies [22, 30]. Thus, recognising carcass species identity in general [27, 28], and studying wild carnivore carcasses in particular [2, 39, 41], is expected to reveal a novel perspective on scavenging behaviour.

This study aims to provide the first insights into the partitioning of carrion of wild mammalian carnivore species amongst the insect and vertebrate scavenger communities, in two Mediterranean mountain areas of south-eastern Spain. A previous experiment carried out in one of those areas [41] showed that the vertebrate scavenging is substantially higher at herbivore carcasses than at carnivore carcasses. Hence, our null hypothesis is that mammalian carnivores will generally avoid the consumption of carnivore carcasses [2, 39, 41], which would imply longer carcass persistence enabling a diverse insect community to colonise the carrion. When discussing our results, we will consider several factors potentially affecting the observed scavenger community dynamics. This includes season and environmental variables, such as precipitation and temperature, which may notably influence the scavenging patterns of vertebrates [1, 6, 8, 42] and have a strong effect on insect activity patterns and distribution [20, 43–45].

## Material and methods

### Ethics statement

Carcasses were obtained from the Wildlife Recovery Center of Murcia and from authorised hunts in the study areas. The Government of Murcia authorized the placement of the carcasses used in this study (permission ref.: AUF 2016/0002). To prevent exposure of scavengers to pathogens and lead, we eviscerated the carcasses, removed the area adjacent to the shot, and discarded the presence of *Trichinella* spp. (see below for details). As our study did not involve direct management of living animals, no further ethical approval was needed in accordance with current regulations on animal experimentation (European Directive 63/2010/EU and Spanish Royal Decree 53/2013).

### Study areas

The study was carried out in two mountainous areas in the region of Murcia (south-eastern Spain): Sierra Espuña Regional Park (hereafter, Espuña) and "El Bebedor" estate (hereafter, Bebedor; Fig 1). Espuña covers an area of 17,804 ha within a marked altitudinal gradient (> 1,300 m; highest point at 1,585 m a.s.l.); the annual mean temperature, the average annual daily minimum (TMIN) and maximum temperature (TMAX) and rainfall in 2016 was 17.6˚C (TMIN = 11.6˚C; TMAX = 23.7˚C) and 326.5 mm, respectively. Bebedor covers an area of 1,600 ha within a moderate altitudinal gradient (~500 m; highest point at 1,596 m a.s.l.); the annual mean temperature and rainfall in 2016 was 13.6˚C (TMIN = 8.5˚C; TMAX = 19.2˚C) and 460.4 mm, respectively (source: SIAM, Murcia Agriculture Information System, available at: http://siam.imida.es). The main vegetation type in both areas corresponds to typical Mediterranean forest, predominantly reforested pine, mixed with meso- and supra-Mediterranean scrubland [46].

Obligate vertebrate scavengers are mainly represented by griffon vultures (*Gyps fulvus* (Hablizl, 1783)) with several, though small, breeding colonies in and around Bebedor and the

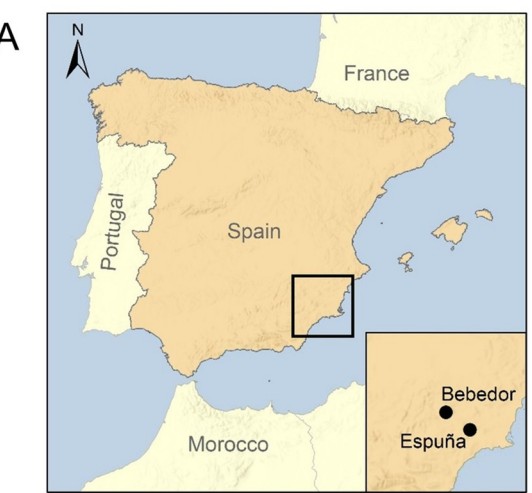

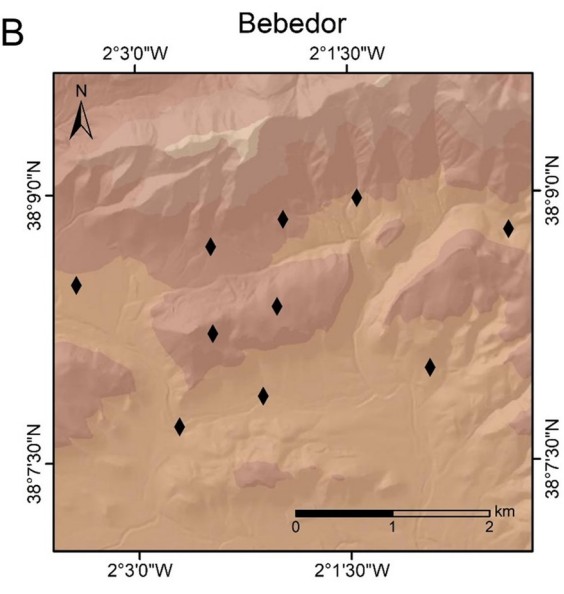

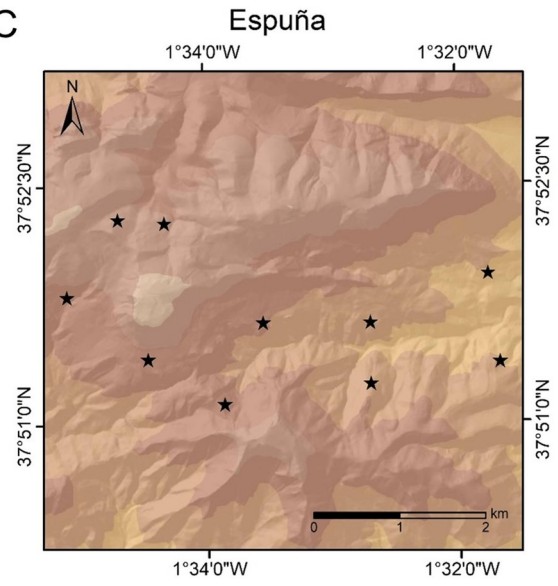

**Fig 1.** Map of continental Spain (a) showing the two study areas: Bebedor (b) and Espuña (c). The locations of monitored carcasses in Bebedor (diamonds) and Espuña (stars) are shown. Maps were generated with ArcGIS 10.1.

presence of non-reproductive individuals in Espuña [32, 47]. The composition of the facultative vertebrate scavenging fauna is also very similar in both study areas. The red fox, *Vulpes vulpes* (Linnaeus, 1758), is the main facultative scavenger in the region [32], but other frequent mammalian and avian scavengers include stone martens, *Martes foina* (Erxleben, 1777), wild boars, *Sus scrofa* Linnaeus, 1758, golden eagles, *Aquila chrysaetos* (Linnaeus, 1758), common ravens, *Corvus corax* Linnaeus, 1758, and magpies, *Pica pica* (Linnaeus, 1758) [9, 32]. Among the insect scavenger fauna, blow flies (Diptera: Calliphoridae) are the main group colonising carrion in Espuña [22]. Several Coleoptera species of the families Dermestidae, Histeridae and Silphidae frequently associated with carrion in Mediterranean forests [14, 43] can also be found in Espuña [22].

## Carcass type and data collection

The red fox was selected as a carnivore carcass model due to its abundance in the two study areas [32, 41] and, thus, it is likely the most frequent carnivore carcass type available in the region under normal circumstances. Red fox was also the main carnivore carcass model used in a previous study that compared the scavenging pattern of carnivore and herbivore carcasses by vertebrate scavengers [41]. Twenty red fox carcasses (10 carcasses for each study area) were monitored from February 2016 to April 2016, i.e., mid-winter to early spring. The carcasses were necropsied and eviscerated prior to storage at -20°C in the Faculty of Veterinary of the University of Murcia. Muscle samples were analysed by artificial digestion to discard the presence of *Trichinella* spp. [48, 49]. Prior to their placement in the study areas, the carcasses were defrosted for 12–24 hours at room temperature.

Each experimental carcass was attached to rocks or trees by a wire, allowing access to both insect and vertebrate scavengers. We always avoided exposing the open part of carcasses to minimize the effect of evisceration on scavenging patterns. Carcasses were randomly placed in southern-oriented sites with intermediate vegetation cover [42], within an altitudinal range of 1000–1400 m a.s.l. Inter-carcass distance was greater than 600 m in every case (Fig 1 and S1 Table), thus ensuring independence of carrion insect succession patterns [50, 51]. One automatic camera (Bushnell Trail Scout and Bushnell Trophy Cam) was placed 5 m away from each carcass to monitor vertebrate scavenger activity. Cameras were set to take one image every minute provided that movement was detected [41] and were active for one week after carcass placement. Due to the failure of one camera in Espuña, vertebrate scavenger activity was monitored at this location by recording any indirect sign of presence, such as tracks and faeces within a radius of 5–10 m around the carcass.

To study carcass use by the community of necrophagous and necrophilous insects, each carcass was visited four times: 3, 7, 29–35 and 56–65 days after carcass placement. The number of visits was kept to a minimum in order to minimise disturbances to any potential vertebrate scavenging activity but, at the same time, visits were distributed over time to document the main stages of decomposition (fresh/bloated, active decay, advanced decay and dry remains) and the insect fauna associated with each stage [22]. Carcasses were always visited around noon to coincide with the daily period of maximum insect activity. During each visit, the relative abundance of each insect family observed on the carcass was visually estimated and recorded for those insects that use carrion itself as a habitat (i.e., necrophagous Diptera larvae; necrophagous and necrophilous Coleoptera adults and larvae). Moreover, a small sample of the different insects was manually collected for further identification in the laboratory (see

below). Relative abundance was estimated using the following five categories: A0 (no observed specimens), A1 (one single observed specimen), A2 (2–10 observed specimens), A3 (11–100 observed specimens) and A4 (more than 100 observed specimens). For those insects that typically visit carrion but do not use it as a habitat (necrophagous Diptera adults that visit carrion to oviposit, and omnivorous and parasitoid Hymenoptera), the observed specimens were collected but their relative abundance was not considered, as their numbers and presence can be highly variable depending on the time of day and weather conditions. During the inspection of each carcass, flying insects perched on the carcass were trapped first; then, the insects on, inside and below the carcass were collected, including representative samples from each egg cluster and larval mass [30, 52]. Post-feeding Diptera larvae were also collected during their dispersal around the carcasses. Collected adult insects and Coleoptera larvae were frozen prior to preservation and storage in 70% ethanol, whereas Diptera eggs and larvae were fixed by immersion in hot water for ~30 seconds prior to preservation and storage in 70% ethanol to prevent decomposition of tissues and facilitate identification, in accordance with published standards and guidelines [53]. Species identification of the insect material was carried out on the basis of diagnostic morphological characters following available keys [54–58].

## Data analyses

For vertebrate scavengers, three variables were considered: 'richness *a*' (number of vertebrate scavenger species visiting the carcass), 'richness *b*' (number of vertebrate scavenger species consuming the carcass), and carcass 'detection time' (time elapsed since the carcass was available until the arrival of the first scavenger, measured in hours) [41]. A vertebrate scavenger was considered a consumer when it was photographed tearing up the carcass and the resulting meat portion was ingested or held in the mouth/beak. We considered scavengers all the carrion consumers detected in this study and those species that had previously been found to scavenge in the study areas [59]. Differences in 'richness *a*', 'richness *b*' and carcass 'detection time' between the two study areas were tested using Mann-Whitney non-parametric tests with post hoc Bonferroni corrections.

For insect scavengers, the factors potentially affecting the richness of necrophagous and necrophilous families per carcass were investigated by fitting generalised linear mixed models (GLMMs) [60]. We also used GLMMs to analyse the factors influencing the abundance of those families observed on at least 25% (n = 20 visits) of the total visits (n = 80). Each family was analysed separately. Thus, 'richness' (no. of families) and the 'abundance' of the more frequent families were the response variables, whereas 'area' (Espuña and Bebedor), 'visit' (first to fourth visit to each carcass) and 'carcass' (as a random factor) were the categorical predictors considered. For each response variable, we constructed a full model including all the variables and their interactions. First, a set of alternative models was constructed using different combinations of the random structure (including a null model, i.e., without a random term) while maintaining the same fixed structure. The model with the most appropriate random structure was then selected using a restricted maximum likelihood (REML) method, with Gaussian error distributions and identity link functions. Second, given that the selected model was the null model in all cases, we used generalised linear models (GLMs) to select the model with the best fixed structure amongst the complete set of model permutations of fixed and interaction terms. We used Akaike's information criterion for small samples sizes (AICc) to identify the most parsimonious model (the one with the lowest AICc) and rank the remaining models, calculating ΔAICc as the difference in AICc between each model and the best model in the set. Those models with ΔAICc < 2 were retained as candidate models, as they are considered to have similar support [58]. When we retained at least two candidate models, we evaluated their

overall degree of support by calculating their proportion of explained deviance ($D^2$) using the following formula [61]: $D^2$ = (null deviance–residual deviance) / null deviance * 100. Moreover, for those candidate models including the explanatory variable 'visit', we performed *post hoc* multiple comparisons of means [62] to explore which visits determined the differences. Finally, an abundance-based correlation matrix was generated to identify potential associations between those families selected for GLMMs (significance set at $p < 0.05$). All the statistical analyses were performed in R 3.0.2 (R Core Team 2013).

## Results

### Vertebrate scavengers

In Espuña, we obtained 22 photographs of 5 vertebrate facultative scavenger species visiting the red fox carcasses (richness *a*: average ± SD = 1.33 ± 0.71): red fox (recorded at 4 carcasses), stone marten (3 carcasses), wild boar (3 carcasses), wildcat, *Felis silvestris* Schreber, 1775 (1 carcass), and golden eagle (1 carcass). Eight of nine carcasses (89%; there was a failure with the camera associated with one of the carcasses) were visited by at least one vertebrate facultative scavenger species. We did not record any indirect signs of vertebrate scavenger activity around the carcass with the non-working camera during the first week after carcass placement. The golden eagle was the only species that partially consumed one of the carcasses (richness *b*: average ± SD = 0.11 ± 0.33). This bird was recorded consuming part of the neck, thorax and forelegs of the fox carcass on days 4, 5 and 6. No other vertebrate species were recorded to contact physically with carcasses.

In Bebedor, we obtained 20 photographs of three vertebrate facultative scavenger species visiting the red fox carcasses (richness *a*: average ± SD = 0.40 ± 0.97): wild boar (recorded at 2 carcasses), red fox (1 carcass) and magpie (1 carcass). Two of ten carcasses (20%) were visited by at least one vertebrate facultative scavenger species; nevertheless, none of them consumed carrion judging by the pictures and the examination of the carcasses during each direct inspection (richness *b*: average = 0). No vertebrate species were recorded to contact physically with carcasses, except a group of four wild boar piglets that probably touched one carcass. In this case, however, the potential contact was not sufficient to move the carcass.

The average carcass detection time (in hours) was similar (W = 27; $p = 0.41$) in both study areas (Espuña: 66.22 ± 20.18; Bebedor: 70.54 ± 51.77). The total number of vertebrate scavenger species recorded in the two study areas was six (four mammals and two birds), with the red fox and the wild boar being detected in both areas. No vultures were recorded at any carcass. Significant differences were found in richness *a* between both study areas (W = 74; $p = 0.01$), with more vertebrate scavenger species visiting carcasses in Espuña. In contrast, no significant differences were found in richness *b* between the two study areas (W = 36; $p = 0.33$).

### Insect scavengers

In total, 19 insect families belonging to orders Diptera, Coleoptera and Hymenoptera were recorded in association with the red fox carcasses (S2 Table). The taxonomic identification of all the recorded insects was confirmed at the laboratory using specimens collected from the carcasses. Among Diptera, the Calliphoridae, specifically the blow fly species *Calliphora vicina* Robineau-Desvoidy, 1830 and *Calliphora vomitoria* (Linnaeus, 1758), were clearly predominant, colonising all the experimental carcasses (Fig 2). Calliphoridae was the only family of Diptera observed in at least 20 of the 80 visits, and thus included in the abundance analysis (see below). Although adults of other sarcosaprophagous Diptera families were collected at carcasses in the two areas during different visits (Fig 3), their eggs and larvae were not observed. It should be noted that lesser dung fly (family Sphaeroceridae) adults occurred in the four visits to the

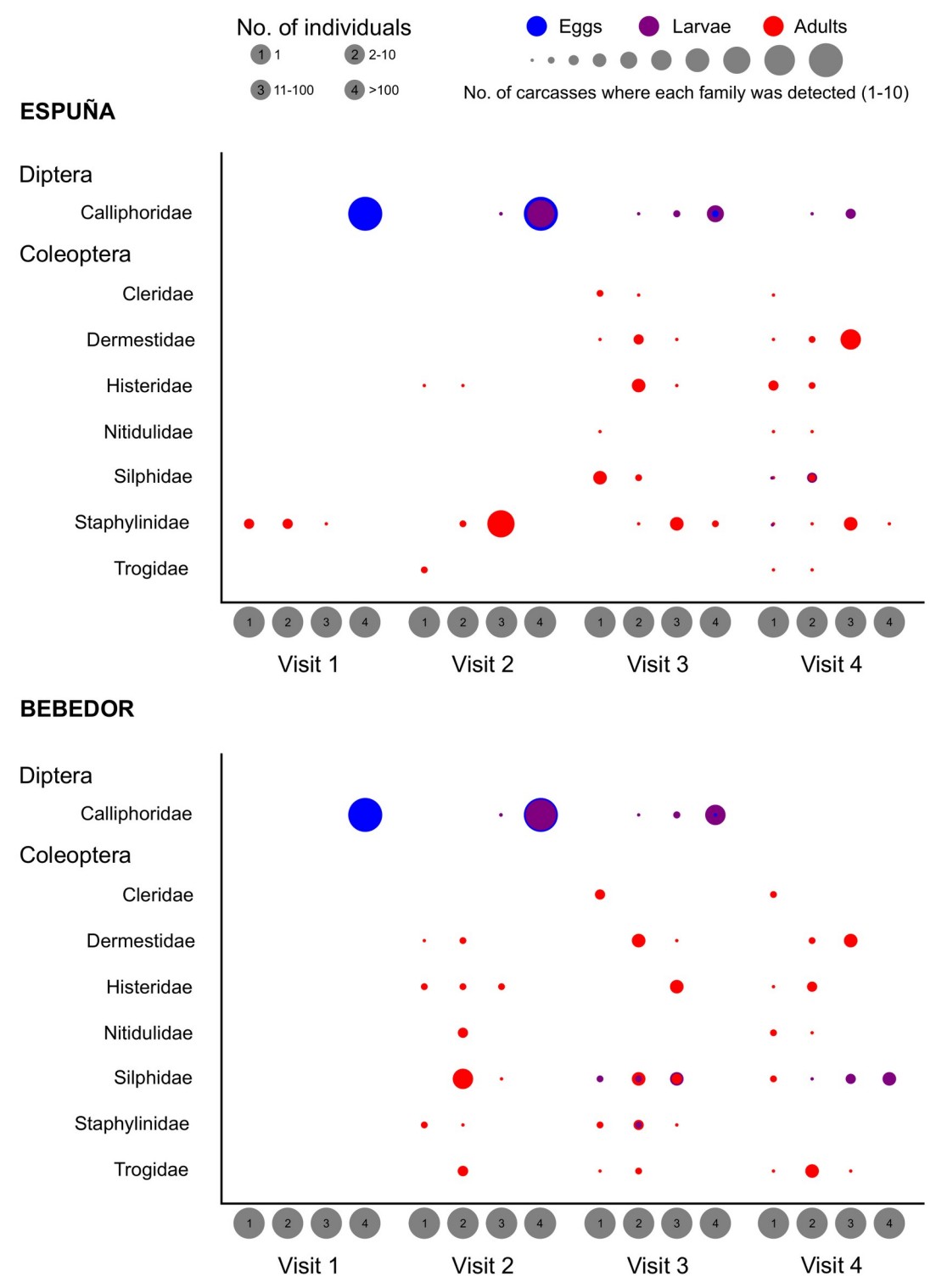

**Fig 2. Frequency of occurrence and relative abundance at red fox carcasses of the most frequently recorded Diptera and Coleoptera families.** Adults of Calliphoridae are not included (see main text).

carcasses in Espuña, but were not observed in Bebedor. Other differences between areas regarding Diptera families were the presence of Heleomyzidae, Phoridae and Sciaridae only in Espuña, and the presence of Fanniidae and Sarcophagidae only in Bebedor (Fig 3).

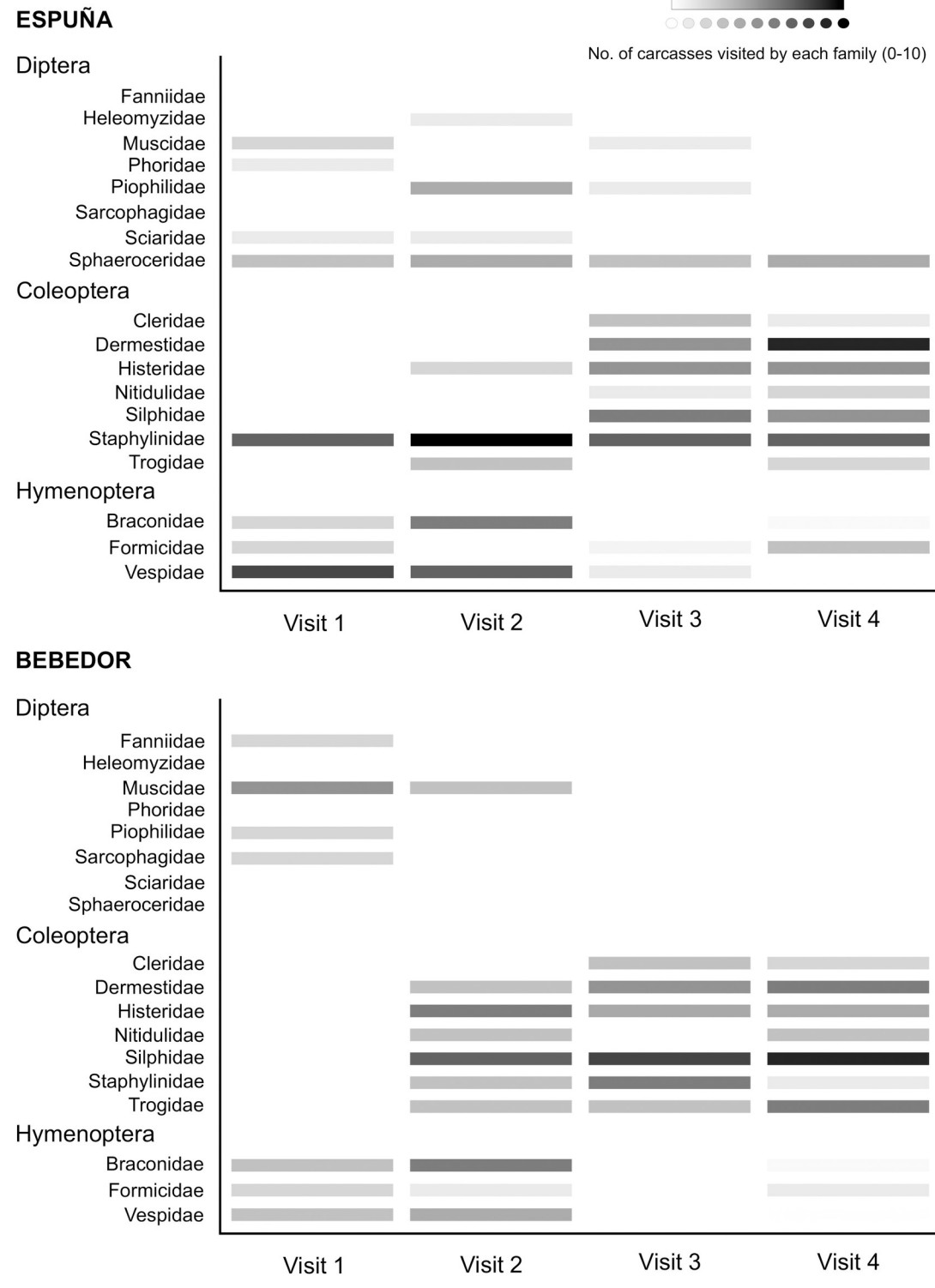

**Fig 3. Frequency of occurrence at red fox carcasses of the insect families for which relative abundance was not estimated.**

Among Coleoptera, seven families were observed in association with carcasses in both study areas (Fig 2), although only the families Dermestidae, Histeridae, Silphidae and Staphylinidae were recorded in at least 20 of the 80 visits, and therefore they were the only ones included in subsequent analyses (see below). Dermestidae and Silphidae were mainly represented by the species *Dermestes frischii* Kugelann, 1792 and *Thanatophilus ruficornis* (Kuster, 1851), respectively, although individuals of *Dermestes undulatus* Brahm, 1790 and *Thanatophilus rugosus* (Linnaeus, 1758) were also relatively frequent (S2 Table). Histeridae was represented by two species, *Saprinus detersus* (Illiger, 1807) and *Saprinus furvus* Erichson, 1834. Staphylinidae was represented by the rove beetle *Creophilus maxillosus* (Linnaeus, 1758) and by several unidentified Aleocharinae. The higher frequency and relative abundance of Staphylinidae beetles in Espuña in comparison to Bebedor is notable (Fig 2). Although not included in subsequent analyses, it should be mentioned that each of the remaining Coleoptera families were represented by one single species: *Necrobia violacea* (Linnaeus, 1758) (Cleridae), *Nitidula flavomaculata* Rossi, 1790 (Nitidulidae) and *Trox perlatus hispanicus* Harold, 1872 (Trogidae).

Finally, the order Hymenoptera was represented in both study areas by adults of the common wasp *Vespula vulgaris* (Linnaeus, 1758) (family Vespidae), the parasitoid wasp *Alysia manducator* (Panzer, 1799) (family Braconidae) and several unidentified ants (family Formicidae). Each of these families was recorded in <20 of the 80 visits, with *V. vulgaris* being more frequently collected in Espuña (Fig 3).

According to our GLM analyses, the observed richness of insect families per carcass was strongly influenced by the visit. The main, significant differences were found between the first and second visits (z = 3.634, p = 0.002; p>0.05 otherwise), when the lowest and highest numbers of families per carcass were recorded, respectively (Fig 4 and Table 1). During the first visit (3 days after carcass placement), carcasses were still in a fresh state of decomposition, with unhatched blow fly eggs as the main specimens observed on carrion. During the second visit (7 days after carcass placement), carcasses were already in an active decay stage of decomposition, as blow fly larvae were consuming the soft tissues in every carcass. At this stage, carcasses appeared to still be suitable for colonisation by blow flies, as new egg batches were observed on every carcass. In Bebedor, all Coleoptera families were first detected during the second visit, with the exception of Cleridae, whose first appearance was recorded during the third visit (29–35 days after carcass placement). In Espuña, most Coleoptera families arrived later, during the third visit (Fig 2). Necrophagous Dermestidae and Silphidae beetles were observed for the first time during the second visit, although only on some carcasses in Bebedor (S2 Table). Overall, insect family richness was slightly higher in Espuña (Figs 2 and 3, Table 2). Nevertheless, the variable area was only represented in the selected models in combination with the variable visit, and the addition of area did not notably increase the explained deviance (Table 1).

The visit was also important in explaining the relative abundance of all insect families, especially Calliphoridae, Dermestidae and Histeridae (Table 1). The abundance of Calliphoridae decreased progressively after the second visit (Fig 2, Table 2), with significant differences between every visit (z = [-11.716]–[-3.226], p<0.01 in all cases) except between the first and the second (p>0.05). The Dermestidae were absent during the first visit to the carcasses but their abundance increased in subsequent visits (Fig 2, Table 2), with significant differences between every visit (z = 2.764–6.738, p<0.05 in all cases) except between the first and the second (p>0.05). The Histeridae peaked in abundance during the third visit to the carcasses (Fig 2, Table 2), with significant differences only found between this and the first visit (z = 3.776, p<0.001; p>0.05 otherwise). As occurred with family richness, the variable area only contributed to explaining (moderately) the abundance of Calliphoridae and Histeridae, either in combination or through its interaction with the visit variable (Table 1).

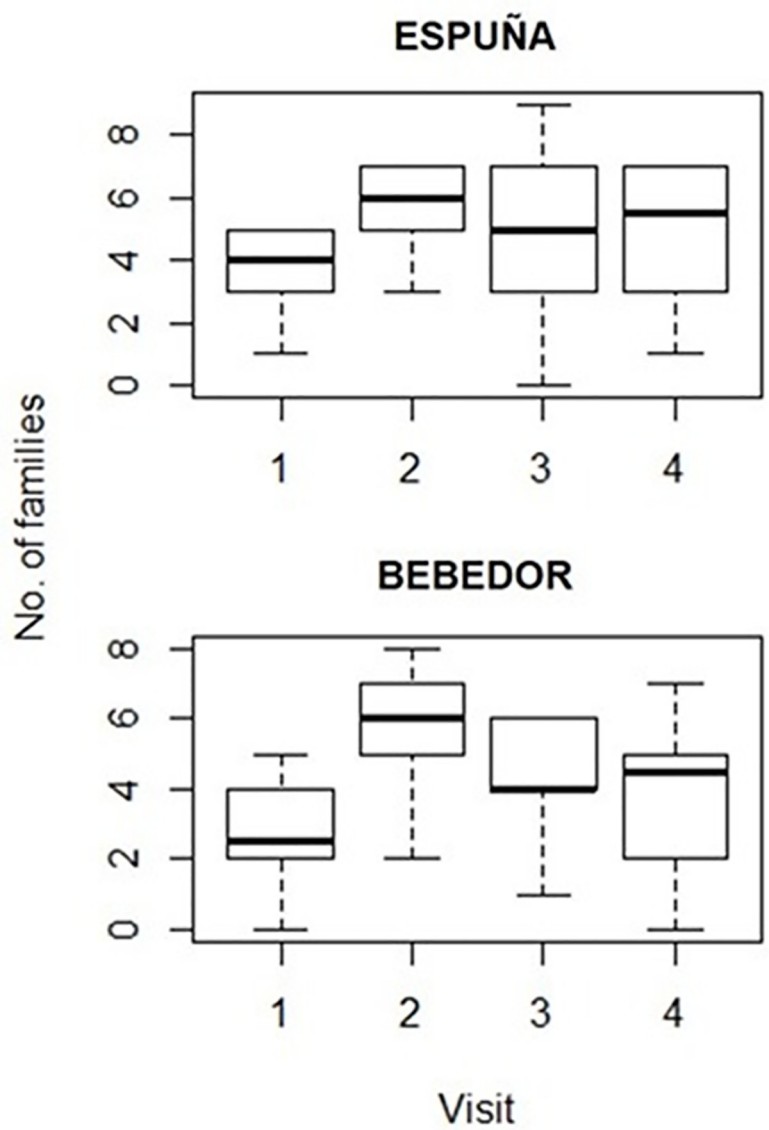

**Fig 4. Number of insect families recorded per visit in Espuña and Bebedor.** The median, the 25% and 75% quartiles and the maximum and minimum values are represented.

The best models explaining the abundance of Silphidae and Staphylinidae included both area and visit (Table 1). The Silphidae were more abundant in Espuña than in Bebedor, although their abundance increased in successive visits to the carcasses in both study areas (Fig 2, Table 2). Significant differences in the observed abundances were found between the first and third visits (z = 4.222, p<0.001), first and fourth visits (z = 5.066, p<0.001) and second and fourth visits (z = 2.955, p<0.0163; p>0.05 otherwise). The Staphylinidae were also more abundant in Espuña, peaking during the second and third visits (Fig 2, Table 2). In this case, significant differences were found only between the first and second visits (z = 3.102, p = 0.011) and first and third visits (z = 3.412, p<0.004; p>0.05 otherwise).

Finally, several significant correlations were found between the abundances of the most frequently collected insect families (Table 3). Positively significant correlations were observed between Dermestidae, Histeridae and Silphidae, as well as between Calliphoridae and

**Table 1. AICc-based model selection to assess the effect of the variables 'area' (Espuña and Nororeste) and 'visit' (1st visit, 2nd visit, 3rd visit and 4th visit to the carcass) on insect family richness and the abundance of those families most frequently recorded on red fox carcasses.** The number of estimated parameters (k), AICc values, AICc differences with the highest-ranked model (ΔAICc), and the percentage of explained deviance ($D^2$) are shown. Selected models are highlighted in bold.

| Response variable | Model | k | AICc | ΔAICc | $D^2$ |
|---|---|---|---|---|---|
| Family richness | **visit** | **3** | **342.36** | **0** | **14.60** |
| | **area + visit** | **4** | **343.15** | **0.79** | **16.29** |
| | area * visit | 7 | 349.53 | 7.17 | |
| | area | 1 | 350.26 | 7.9 | |
| Calliphoridae abundance | **visit** | **3** | **222.33** | **0** | **70.73** |
| | **area * visit** | **7** | **222.7** | **0.33** | **73.98** |
| | **area + visit** | **4** | **224.12** | **1.79** | **70.94** |
| | area | 1 | 315.98 | 93.65 | |
| Silphidae abundance | **area * visit** | **7** | **217.48** | **0** | **59.33** |
| | area + visit | 4 | 224.72 | 7.22 | 51.15 |
| | visit | 3 | 252.39 | 34.91 | |
| | area | 1 | 255.08 | 37.6 | |
| Histeridae abundance | **visit** | **3** | **227.73** | **0** | **16.27** |
| | **area + visit** | **4** | **228.64** | **0.91** | **17.75** |
| | area * visit | 7 | 232.7 | 4.97 | |
| | area | 1 | 236.24 | 8.51 | |
| Staphylinidae abundance | **area + visit** | **4** | **238.09** | **0** | **47.12** |
| | **area * visit** | **7** | **238.84** | **0.75** | **51.35** |
| | area | 1 | 246.74 | 8.65 | |
| | visit | 3 | 277.14 | 39.05 | |
| Dermestidae abundance | **visit** | **3** | **219.55** | **0** | **42.24** |
| | area + visit | 4 | 221.88 | 2.33 | |
| | area * visit | 7 | 224.45 | 4.9 | |
| | area | 1 | 258.95 | 39.4 | |

Staphylinidae. Interestingly, both Calliphoridae and Staphylinidae were negatively and significantly correlated with Silphidae. The abundance of Calliphoridae was also negatively and significantly correlated with the abundance of Dermestidae (Table 3).

## Discussion

### Scavenging by vertebrates

As expected, the consumption of red fox carcasses was generally avoided by vertebrate scavengers [2, 39, 41]. We only detected carrion consumption by the golden eagle, despite the fact that carcasses were frequently visited by several mammalian carnivore species including the red fox, which visited the most carcasses. This agrees with previous findings showing that mammalian carnivores deliberately avoid the consumption of carnivore carcasses [41]. As argued by these authors, the avoidance of pathogen transmission risk might be the main reason behind this behaviour, leading to the evolution of a low preference for carcasses of phylogenetically related species and very low cannibalistic tendencies in vertebrate scavengers. Thus, red fox carcasses generally persist for longer periods than herbivore carcasses [41], making them widely available to other organisms.

Contrary to our expectations, vultures did not consume any carcass, which could be explained by the relatively low population density of vultures in our study areas. In a study conducted by our team in a similar mountainous area of south-eastern Spain (Sierra de

**Table 2. Selected generalised lineal models (GLMs) showing the relationship between variables studied ("area" and "visit") and response variables (family richness and abundance of the most frequent families).** The estimate of the parameters (including the sign), the standard error of the parameters (SE) and the degrees of freedom of the models (DF) are shown.

| Response variable | Model | Parameter | Estimate | SE | DF |
|---|---|---|---|---|---|
| Family richness | **visit** | Intercept | 1.163 | 0.125 | 79 |
| | | visit (2) | 0.569 | 0.156 | |
| | | visit (3) | 0.341 | 0.164 | |
| | | visit (4) | 0.319 | 0.164 | |
| | **area + visit** | Intercept | 1.226 | 0.135 | 79 |
| | | area (Bebedor) | -0.130 | 0.106 | |
| | | visit (2) | 0.569 | 0.156 | |
| | | visit (3) | 0.341 | 0.164 | |
| | | visit (4) | 0.319 | 0.164 | |
| Calliphoridae abundance | **visit** | Intercept | 4.000 | 0.208 | 79 |
| | | visit (2) | 3.039e-15 | 0.295 | |
| | | visit (3) | -0.950 | 0.295 | |
| | | visit (4) | -3.450 | 0.295 | |
| | **area * visit** | Intercept | 4.000 | 0.285 | 79 |
| | | area (Bebedor) | -1.316e-15 | 0.404 | |
| | | visit (2) | -5.160e-16 | 0.404 | |
| | | visit (3) | -1.200 | 0.404 | |
| | | visit (4) | -2.900 | 0.404 | |
| | | area (Bebedor) : visit (2) | 2.265e-15 | 0.571 | |
| | | area (Bebedor) : visit (3) | 0.500 | 0.571 | |
| | | area (Bebedor) : visit (4) | 1.100 | 0.571 | |
| | **area + visit** | Intercept | 4.075 | 0.234 | 79 |
| | | area (Bebedor) | -0.150 | 0.209 | |
| | | visit (2) | 9.746e-16 | 0.295 | |
| | | visit (3) | -0.950 | 0.295 | |
| | | visit (4) | -3.450 | 0.295 | |
| Silphidae abundance | **area * visit** | Intercep | -1.287e-15 | 0.276 | 79 |
| | | area (Bebedor) | 3.426e-16 | 0.391 | |
| | | visit (2) | 6.880e-16 | 0.391 | |
| | | visit (3) | 0.800 | 0.391 | |
| | | visit (4) | 0.800 | 0.391 | |
| | | area (Bebedor) : visit (2) | 1.500 | 0.552 | |
| | | area (Bebedor) : visit (3) | 1.400 | 0.552 | |
| | | area (Bebedor) : visit (4) | 2.000 | 0.552 | |
| Histeridae abundance | **visit** | Intercept | -6.703e-16 | 0.215 | 79 |
| | | visit (2) | 0.750 | 0.305 | |
| | | visit (3) | 1.150 | 0.305 | |
| | | visit (4) | 0.700 | 0.305 | |
| | **area + visit** | Intercept | -0.125 | 0.240 | 79 |
| | | area (Bebedor) | 0.250 | 0.215 | |
| | | visit (2) | 0.750 | 0.304 | |
| | | visit (3) | 1.150 | 0.304 | |
| | | visit (4) | 0.700 | 0.304 | |
| Staphylinidae abundance | **area + visit** | Intercept | 1.413 | 0.255 | 79 |
| | | area (Bebedor) | -1.625 | 0.228 | |
| | | visit (2) | 1.000 | 0.322 | |

*(Continued)*

**Table 2.** (Continued)

| Response variable | Model | Parameter | Estimate | SE | DF |
|---|---|---|---|---|---|
| | | visit (3) | 1.100 | 0.322 | |
| | | visit (4) | 0.350 | 0.322 | |
| | **area * visit** | Intercept | 1.200 | 0.316 | 79 |
| | | area (Bebedor) | -1.200 | 0.446 | |
| | | visit (2) | 1.600 | 0.446 | |
| | | visit (3) | 1.000 | 0.446 | |
| | | visit (4) | 0.700 | 0.446 | |
| | | area (Bebedor) : visit (2) | -1.200 | 0.631 | |
| | | area (Bebedor) : visit (3) | 0.200 | 0.631 | |
| | | area (Bebedor) : visit (4) | -0.700 | 0.631 | |
| Dermestidae abundance | **visit** | Intercept | -1.365e-16 | 0.205 | 79 |
| | | visit (2) | 0.250 | 0.289 | |
| | | visit (3) | 1.050 | 0.289 | |
| | | visit (4) | 1.950 | 0.289 | |

Cazorla) during winter, vultures were recorded consuming 33% of red fox carcasses [41]. There, vultures are notably more abundant than in our study areas [32] (authors' unpublished data). Thus, high vulture population densities could influence the potential community of necrophagous and necrophilous insects associated with red fox carcasses by reducing their average persistence period.

## Scavenging by insects

In accordance with our hypothesis, the longer persistence and availability of carnivore carcasses in the study areas enabled the succession of a diverse insect community. Although sampling in our study was non-intensive so as not to disturb the activity of vertebrate scavengers, the patterns and species composition of the observed insect succession were overall in accordance with a previous study involving intensive insect sampling on a piglet carcass in Espuña [22]. In both studies, flies of the family Calliphoridae were always the first colonisers of carrion. Blow fly females are able to locate fresh carcasses promptly after death under different conditions [21, 63]. This ability, together with their high fecundity and short generation times, explains why blow flies generally dominate the carrion insect community during the initial stages of decomposition [10]. The two blow fly species that colonised the carcasses during the present study, *C. vicina* and *C. vomitoria*, are the main active species during the cool seasons in the Iberian Peninsula, when they have been recorded colonising pig carcasses within the first days after death [22, 30, 63]. Other insects, like silphid beetles of the genus *Thanatophilus* Leach, 1815, also feed on soft tissues during the first stages of decomposition, but they are

**Table 3. Correlation matrix for the abundances of the insect families most frequently collected on the red fox carcasses.** Statistically significant correlations (α = 0.05) are highlighted in bold.

| | Calliphoridae | Silphidae | Dermestidae | Staphylinidae | Histeridae |
|---|---|---|---|---|---|
| Calliphoridae | 1 | **-0.395** | **-0.460** | **0.289** | 0.001 |
| Silphidae | | 1 | **0.362** | **-0.256** | **0.463** |
| Dermestidae | | | 1 | 0.072 | **0.454** |
| Staphylinidae | | | | 1 | 0.143 |
| Histeridae | | | | | 1 |

generally secondary colonisers, arriving at carcasses up to two weeks after death [22, 64]. During that period of time, soft tissues are still available if carcasses are protected from vertebrate scavenging [22]. However, in natural conditions, non-carnivore carcasses in the Mediterranean and other warm environments can be completely consumed by vertebrate scavengers in a few hours or days [31, 32, 59, 65]. This potentially affects those insects, like silphid beetles, that might require more time to detect and colonise carrion [41]. Moreover, in areas with intense competition between vultures and facultative vertebrate scavengers, insects consume only a small amount of carrion from herbivore carcasses [59, 66], so the first colonisers may have a clear advantage. All of this may explain why, in our study, the abundance of silphid beetles was influenced by both the area and the visit (Table 1), being lower in Bebedor and increasing with successive visits in both study areas, whereas the abundance of blow flies was similar in both areas but decreased after the second visit. In addition, the abundances of blow flies and silphid beetles were negatively and significantly correlated (Table 3). Interestingly, carrion insect succession studies from central [67] and southern Europe [30, 68] have shown that, under prolonged cold weather conditions, blow flies may be delayed and silphid beetles could then be the first colonisers of carcasses. Those situations result in atypical cases where larvae of silphid beetles drive the active decomposition of soft tissues and dominate the insect community over blow flies [67]. This suggests that there might be intense competitive dynamics between blow flies and silphid beetles that are not yet fully understood. For example, it has been noted that silphid beetles show different seasonal patterns between different bioclimatic levels in Mediterranean habitats, apparently to avoid direct competition with blow flies [43].

As in the case of the Silphidae, the abundance of beetles of the family Staphylinidae was influenced by both the area and the visit (Table 1). Staphylinids were more abundant in Espuña, an area where carcasses may be more readily available due to both the scarcity of vultures and the regular supply of ungulate carcasses through sport hunting and culling practices [69]. Many staphylinid species are actually necrophilous; some of them are predators of blow fly larvae whereas species within the subfamily Aleocharinae are parasitoids of blow fly pupae [13]. It is therefore not surprising that the abundances of blow flies and staphylinid beetles were positively and significantly correlated (Table 3). The staphylinids showed a higher abundance during the second and third visits, when the density of blow fly larvae feeding on carcasses was higher and, therefore, so was the likelihood of finding post-feeding larvae ready to pupariate. Similarly, the abundance of beetles of the family Histeridae was higher during the third visit, coinciding with high densities of blow fly larvae. Many histerid species are indeed predators of blow fly larvae [14], although no correlation was found between their abundance and that of blow flies (Table 3). Histerid beetles are mainly active during the warm seasons [14], which may explain why their abundance in the second visits to the carcasses (performed in February) was not higher. On the other hand, the two histerid species collected during our study, *S. detersus* and *S. furvus*, are widely distributed among different Mediterranean habitats [14] and, accordingly, their abundance was not influenced by the area. However, necrophilous histerid beetles are highly diverse in Mediterranean areas and several species show a strong preference for certain habitats [14].

Whereas blow flies typically colonise fresh carcasses to feed on soft tissues, beetles of the family Dermestidae often succeed them during advanced stages of decomposition to consume dry tissues [43]. Hence, the observed negative and significant correlation between blow fly and dermestid abundances (Table 3) is not surprising, nor is the increasing abundance of dermestids in subsequent visits. The two dermestid species most frequently collected during our study, *D. frischii* and *D. undulatus*, are widely distributed throughout different Mediterranean habitats [43] and commonly associated with human remains in forensic contexts [70, 71]. Most insect species collected during our study have indeed been collected not only from other

vertebrate carcasses, including humans, but also using invertebrate carrion-baited traps in Mediterranean habitats [14, 20, 22, 43, 44, 68, 72, 73]. This suggests that most necrophagous insect species are generalists and colonise most types of vertebrate carrion. Watson and Carlton [27, 28] found, however, that, when access to carcasses by vertebrate scavengers is experimentally prevented, the insect communities associated with carcasses of three mammal species (black bear, *Ursus americanus* Pallas, 1780, white-tailed deer, *Odocoileus virginianus* (Zimmermann, 1780), and swine, *Sus scrofa domestica* Erxleben, 1777) were more diverse than those associated with alligator carcasses, and, within mammal carcass types, insect diversity was higher on white-tailed deer and swine than on black bears.

Whilst the few published studies that had considered the scavenging of carrion by both vertebrates and invertebrates typically used small-sized carcasses, such as mice [24, 25], chicks [6] or rabbits [26], our study provides the first data on the partitioning of medium-sized carnivore carcasses. Under experimental conditions of carnivore scavenger exclusion, carcass size has been identified as a key factor determining the structure of the insect scavenger communities [29]. Similarly to vertebrate scavenger communities [31], small-sized carcasses are often monopolised by a single or a few insect species [74], whereas larger carcasses generally show multi-guild competition patterns of insect-driven decomposition [29]. In fact, our results show a well-structured insect community including not only necrophagous species but also omnivores and necrophilous predators and parasitoids.

## Caveats, future directions and conclusions

We must note that it was not our aim to provide a complete inventory of the insects associated with mammalian carnivore carcasses, but to investigate, for the first time, the partitioning of carnivore carcasses amongst vertebrate and insect scavengers, considering that carnivore carcasses are generally avoided by carnivores [41]. Insect succession studies generally involve daily collections, at least during the first stages of decomposition [22, 64, 67]. As previously mentioned, this was not possible in our study without potentially disturbing vertebrate scavenging activity. In contrast, the number of carcass replicates in our study was considerably higher than in most insect succession surveys, sometimes limited to examining one or two carcasses in one single study area [20, 22, 30, 68]. This, together with the agreement between our findings and a previous study on the carrion insect communities carried out in the same area [22], supports the representativeness of our data. Regardless, more complete records of insect activity at carcasses, without disturbing vertebrate scavenging activity, would be needed to ensure that the absence of particular species is not due to infrequent sampling. In addition, future studies could consider tracking vertebrate scavenger activity with automatic cameras throughout the experiment. In our study, we limited the use of cameras to the first week after carcass placement; nonetheless, visual inspections of the carcasses did not reveal signals of vertebrate consumption in the subsequent weeks, which agrees with Moleón et al. [41], who found very little scavenging activity by vertebrates over 5–9 weeks of monitoring. It should also be noted that we used necropsied and eviscerated red fox carcasses, in order to minimise the risk of pathogen transmission [41]. The potential effect of the use of eviscerated carcasses on the patterns of carrion consumption by both vertebrate and invertebrate scavengers deserves further study. However, several studies have demonstrated that the rate of decomposition and the pattern of insect colonisation do not differ between intact and necropsied and/or wounded carcasses [75, 76]. In this sense, the composition of the insect fauna recorded during our study was overall similar to that reported from non-eviscerated, small- and medium-sized pig carcasses which were protected from vertebrate scavenging [22, 30]. Furthermore, our experimental carcasses were colonised by blow flies within the first three days after carcass

exposure (Fig 2). This is in line with the early colonisation patterns of carrion during winter, when the arrival of blow flies can sometimes be delayed a few days, in contrast to warmer months, when carrion can be colonised within hours after death [30, 63].

Carnivores generally avoid the consumption of carnivore carcasses [41], as evidenced by the fact that only one of our experimental carcasses was partially consumed by a vertebrate scavenger, a golden eagle. Given this, future research should investigate the partitioning of herbivore carcasses and the potential impact of scavenging by vertebrates on the carrion insect community. Interestingly, the carrion insect community had been found to be more diverse on herbivore and omnivore carcasses than on carnivore carcasses if the access of vertebrate scavengers is experimentally prevented [27, 28]. However, under natural conditions the carrion insect communities associated with herbivore carcasses may actually be less diverse than those on carnivore carcasses if the latter are avoided by vertebrate facultative scavengers and persist longer in the field [41]. A medium- or large-sized herbivore carcass can be completely consumed by vertebrate scavengers within a few hours or days [32], which contrasts with the persistence of soft tissues in small-sized piglet carcasses protected from vertebrate scavengers up to two weeks after carcass exposure [22].

Some necrophagous insects may require previous scavenging by vertebrates to colonise specific tissues of a carcass; for example, certain flies of the family Piophilidae, commonly called 'bone-skippers', may need vertebrate scavengers to act on ungulate carcasses first in order to access to the bone marrow, their preferential breeding site [73]. While there are several species of bone-skippers distributed throughout the Mediterranean region, they are absent from insect succession studies where carcasses are protected from vertebrate scavenging and have typically been recorded in areas where vultures are present [73, 77]. As indirect regulators of vertebrate facultative scavenger populations [26, 32, 65], vultures may also have a major impact on carrion insect communities. Future studies performed in areas with higher vulture densities will provide further insights into the carrion partitioning amongst insect and vertebrate scavengers.

Given the seasonal changes in species composition shown by carrion insect communities in temperate regions [43–45], future studies on carrion partitioning between vertebrates and invertebrates could examine the effects of different seasons. Studies in tropical ecosystems are also especially interesting. In Afrotropical forests, a high level of fly larvae activity at carcasses may outcompete vertebrate scavengers including vultures [33]. This contrasts with Neotropical forests, where a complex community of ants suppresses maggot infestations on carcasses, thereby extending the availability of carcasses to vertebrates [34]. Ultimately, sound baseline data on the successional patterns of those insect communities associated with different wild animal carcasses that are accessible to vertebrate scavengers will enhance wildlife forensic investigations, enabling the reconstruction of taphonomic processes and the estimation of a minimum post-mortem interval based on insect activity [19, 78, 79].

In conclusion, this is the first study analysing the scavenging of carnivore carrion by both vertebrates and invertebrates. Our study provides additional support to the carnivore carrion avoidance hypothesis, which suggests that mammalian carnivores avoid the consumption of carnivore carcasses, especially of conspecifics [41]. Moleón et al. [41] also suggested that the absence of vertebrate scavengers at carnivore carcasses increases the amount of carrion biomass available to other organisms, thus enabling a successional insect community to colonise the carcasses. This was confirmed in our study, in which we found a well-structured, diverse insect community of necrophages, omnivores and necrophilous predators and parasitoids comparable to those insect communities reported in succession studies where access of vertebrate scavengers to herbivore and omnivore carrion was prevented [22, 30]. Carrion resource partitioning amongst invertebrate and vertebrate scavengers in natural conditions represents a

virtually unexplored avenue for future research that may provide promising, holistic insights into carrion ecology.

## Supporting information

**S1 Table. GPS coordinates of carcasses.**
(DOCX)

**S2 Table. List of insect taxa collected during each visit in the two study areas.**
(DOCX)

## Acknowledgments

Enemérito Muñiz and Fernando Escribano helped during fieldwork. Dirección General del Medio Natural (Consejería de Agua, Agricultura y Medio Ambiente. Comunidad Autónoma de la Región de Murcia) and Sierra Espuña Regional Park provided help with logistics and permissions. Stefano Vanin and an anonymous reviewer provided useful comments and suggestions on the present manuscript. D.M.-V. was supported by an EC funded Marie Curie Intra-European Fellowship (FP7-PEOPLE-2013-IEF-624575) and a research contract from the University of Alcalá (Ayudas Postdoctorales UAH), Z.M.-R. by a pre-doctoral grant (FPU12/00823), and M.M. by a research contract Ramón y Cajal from the MINECO (RYC-2015-19231). This study was partly funded by the Spanish Ministry of Economy, Industry and Competitiveness and EU ERDF funds through the projects CGL2015-66966-C2-1-2-R and CGL2017-89905-R.

## Author Contributions

**Conceptualization:** Carlos Muñoz-Lozano, Daniel Martín-Vega, Carlos Martínez-Carrasco, José A. Sánchez-Zapata, Marcos Moleón.

**Formal analysis:** Carlos Muñoz-Lozano, Marcos Moleón.

**Funding acquisition:** Marcos Moleón.

**Investigation:** Carlos Muñoz-Lozano, Daniel Martín-Vega, Carlos Martínez-Carrasco, José A. Sánchez-Zapata, Zebensui Morales-Reyes, Moisés Gonzálvez, Marcos Moleón.

**Methodology:** Carlos Muñoz-Lozano, Daniel Martín-Vega, Carlos Martínez-Carrasco, José A. Sánchez-Zapata, Zebensui Morales-Reyes, Moisés Gonzálvez, Marcos Moleón.

**Project administration:** Marcos Moleón.

**Writing – original draft:** Carlos Muñoz-Lozano, Daniel Martín-Vega, Carlos Martínez-Carrasco, Marcos Moleón.

**Writing – review & editing:** Carlos Muñoz-Lozano, Daniel Martín-Vega, Carlos Martínez-Carrasco, José A. Sánchez-Zapata, Zebensui Morales-Reyes, Moisés Gonzálvez, Marcos Moleón.

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
