## [Decision Letter · Decision Letter 0]

23 Jul 2019

PONE-D-19-16398

Avoidance of carnivore carcasses by vertebrate scavengers enables colonization by a diverse community of carrion insects

PLOS ONE

Dear Dr. Martin-Vega,

Thank you for submitting your manuscript to PLOS ONE. After careful consideration, we feel that it has merit but does not fully meet PLOS ONE’s publication criteria as it currently stands. Therefore, we invite you to submit a revised version of the manuscript that addresses the points raised during the review process.

We would appreciate receiving your revised manuscript by Sep 06 2019 11:59PM. To enhance the reproducibility of your results, we recommend that if applicable you deposit your laboratory protocols in protocols.io, where a protocol can be assigned its own identifier (DOI) such that it can be cited independently in the future. For instructions see: http://journals.plos.org/plosone/s/submission-guidelines#loc-laboratory-protocols

We look forward to receiving your revised manuscript.

Kind regards,

Pierfilippo Cerretti, Ph.D.

Academic Editor

PLOS ONE

Journal Requirements:

1. Thank you for including your ethics statement: All applicable institutional and/or national guidelines for the care and use of animals were followed.

Please amend your current ethics statement to include the full name of the ethics committee that approved your specific study.

For additional information about PLOS ONE submissions requirements for animal ethics, please refer to http://journals.plos.org/plosone/s/submission-guidelines#loc-animal-research  

Additional Editor Comments (if provided):

Dear Daniel,

Two reviewers provided comments and suggestions to your manuscript. They are both positive with your work, so I encourage you to resubmit a revised version according to their comments. Should you desagree with some suggestions, please prepare and submitt a point-by-point rebuttal to the relevant comments, in a separate file.

Best wishes,

Pierfilippo Cerretti

Reviewers' comments:

Reviewer's Responses to Questions

**Comments to the Author**

1. Is the manuscript technically sound, and do the data support the conclusions?

Reviewer #1: Yes

Reviewer #2: Yes

2. Has the statistical analysis been performed appropriately and rigorously? 

Reviewer #1: Yes

Reviewer #2: Yes

3. Have the authors made all data underlying the findings in their manuscript fully available?

Reviewer #1: Yes

Reviewer #2: No

4. Is the manuscript presented in an intelligible fashion and written in standard English?

Reviewer #1: Yes

Reviewer #2: Yes

5. Review Comments to the Author

Reviewer #1: Dear Editor, the paper PONE-D-19-16398 by Munoz-Lozano and co-workers deals with a very important topic in forensic entomology especially when wildlife is considered.

experimental design, number of replicates and the statisics applied are sound and very well developed.

Only few minor observations to improve the text:

General

Add the Authority and the year of description of all the species (vertebrates as well) when reported the first time

Introduction

-add a sentence indicating that as well some work on microbiome of wild animals has been performed (eg.: Tuccia F, Zurgani E, Bortolini S, Vanin S Experimental evaluation on the applicability of necrobiome analysis in forensic veterinary science. Microbiologyopen. 2019 Mar 12:e828. doi: 10.1002/mbo3.828)

line 37, mention as well the very important book published by M. Eric Benbow, Jeffery K. Tomberlin, Aaron M. Tarone in 2018 (Carrion Ecology, Evolution, and Their Applications)

line 38 add "and veterinary" after medicolegal

line 51 add "or coancealed" after [2]

line 55 affect may be better than condition

line 58 add "being" at the end of the line

line 62 delete "Forensic" at the beginning of the sentence

line 63 add "of" after surrogate

M&M

-add a space between the degree symbol and C

line 100 add as well the max and minimum temperature

line 131 bette 600 m than 0.6 Km...in the whole paragraph you are using m as unit

lines 200 and 203 check the way in which the references are cited (apex)

Results

line 272 vulgaris not vularis

line 352 change similar with related

line429 add to the reference 67 as well Bugelli et al., Decomposition pattern and insect colonization in two cases of suicide by hanging Forensic Sci Res. 2018; 3(1): 94–102. doi: 10.1080/20961790.2017.1418622

Reviewer #2: It is an interesting manuscript related with carrion ecology. Contribution is original and worthy of publication in PLoS ONE. The authors discuss about most of the weaknesses of experiment. I would like to suggest some additional corrections.

1. Could provide (maybe in appendix) detailed data (GPS coordinates) for particular exposed experimental carrions? Small scale differences in carrion exposition may have very strong impact on insect succession. Sometimes it is stronger than between distant locations but where carrions are exactly in the same habitats and with the same abiotic conditions.

2. Experimental fox carrion was not eaten by vertebrate scavengers (except golden eagle). However authors noticed many incidents of visiting of experimental carrion by many vertebrate species. Could you describe this contact with more details? Has any physical contact with carrion occurred (jerking, moving etc.)?

3. I have some hesitation related to quite low number of repetitions in the experiment. The information about detailed position of particular experimental carrions (suggested above) may tell us that material from each carrion is really comparable to others.

4. I’m not sure (in contrary to authors…) that evisceration of carrion is without serious influence on the insect succession. Did authors do any tests for comparison of insect succession for eviscerated vs. eviscerated fox carrion? If not, it should be mentioned in the text.

6. PLOS authors have the option to publish the peer review history of their article (what does this mean?). If published, this will include your full peer review and any attached files.

Reviewer #1: Yes: Stefano Vanin

Reviewer #2: No

---

## [Author Response · Author response to Decision Letter 0]

1 Aug 2019

Dear Editor,

We are submitting a revised version of the manuscript entitled “Avoidance of carnivore carcasses by vertebrate scavengers enables colonization by a diverse community of carrion insects” (manuscript ID PONE-D-19-16398), modified in accordance with the comments and suggestions from the reviewers. The changes made in the manuscript are detailed below, together with the responses to the correspondent reviewer’s comments. We are grateful to the two reviewers for their helpful and constructive recommendations and overall positive feedback, and we hope that the manuscript can be accepted for publication in PLOS ONE. 

Yours sincerely,

Daniel Martín-Vega

Reviewers' comments:

Reviewer #1: 

Dear Editor, the paper PONE-D-19-16398 by Munoz-Lozano and co-workers deals with a very important topic in forensic entomology especially when wildlife is considered.

experimental design, number of replicates and the statisics applied are sound and very well developed.

Only few minor observations to improve the text:

General

Add the Authority and the year of description of all the species (vertebrates as well) when reported the first time

*Our response: The authority and year of description have been added for all the species when mentioned in the main text for the first time, as well as on supplementary Table S2 (“Table S1” in the previous version of the manuscript).

Introduction

-add a sentence indicating that as well some work on microbiome of wild animals has been performed (eg.: Tuccia F, Zurgani E, Bortolini S, Vanin S Experimental evaluation on the applicability of necrobiome analysis in forensic veterinary science. Microbiologyopen. 2019 Mar 12:e828. doi: 10.1002/mbo3.828)

*Our response: The following sentence with the suggested reference has been included: “In a similar way, some studies have also explored the potential of the changes in the bacterial communities during the decomposition process of animal carcasses as indicators for death time estimations in forensic veterinary science [23]” (lines 43–45 of the new version of the manuscript).

line 37, mention as well the very important book published by M. Eric Benbow, Jeffery K. Tomberlin, Aaron M. Tarone in 2018 (Carrion Ecology, Evolution, and Their Applications)

*Our response: The suggested reference has been included (new reference number [17]) on line 37, as well as the reference of a book recently published which is also relevant to the present paper (Olea PP, Mateo-Tomás P, Sánchez-Zapata JA, editors. Carrion Ecology and Management. Dordrecht: Springer, 2019; new reference number [18]). All the reference numbers have been revised and updated accordingly in the new version of the manuscript.

line 38 add "and veterinary" after medicolegal

*Our response: Added as suggested.

line 51 add "or coancealed" after [2]

*Our response: Added as suggested (line 54 of the new version of the manuscript).

line 55 affect may be better than condition

*Our response: “condition” has been changed to “affect” (line 58 of the new version of the manuscript)

line 58 add "being" at the end of the line

*Our response: Added as suggested (line 62 of the new version of the manuscript).

line 62 delete "Forensic" at the beginning of the sentence

*Our response: “Forensic” has been deleted as suggested (line 65 of the new version of the manuscript).

line 63 add "of" after surrogate

*Our response: Added as suggested (line 66 of the new version of the manuscript).

M&M

-add a space between the degree symbol and C

*Our response: Thank you for this suggestion. We have checked style manuals and several previous papers published in PLOS ONE and we have found that the number, degree symbol and capital C are written together. We have left it in this way; nevertheless, we will be happy to change it if another way is preferred by the journal. 

line 100 add as well the max and minimum temperature

*Our response: Maximum and minimum temperatures have been added in lines 113–115 and 117–119 of the new version of the manuscript.

line 131 bette 600 m than 0.6 Km...in the whole paragraph you are using m as unit

*Our response: Changed as requested (line 157 of the new version of the manuscript).

lines 200 and 203 check the way in which the references are cited (apex)

*Our response: Reference style has been amended and the reference number updated (reference 61 in line 226 and reference 620in line 229 of the new version of the manuscript). 

Results

line 272 vulgaris not vularis

*Our response: The species name has been corrected (line 305 of the new version of the manuscript).

line 352 change similar with related

*Our response: Changed as suggested (line 386 of the new version of the manuscript).

line429 add to the reference 67 as well Bugelli et al., Decomposition pattern and insect colonization in two cases of suicide by hanging Forensic Sci Res. 2018; 3(1): 94–102. doi: 10.1080/20961790.2017.1418622

*Our response: The suggested reference (reference [71]) has been added (line 464 of the new version of the manuscript).

Reviewer #2: 

It is an interesting manuscript related with carrion ecology. Contribution is original and worthy of publication in PLoS ONE. The authors discuss about most of the weaknesses of experiment. I would like to suggest some additional corrections.

1. Could provide (maybe in appendix) detailed data (GPS coordinates) for particular exposed experimental carrions? Small scale differences in carrion exposition may have very strong impact on insect succession. Sometimes it is stronger than between distant locations but where carrions are exactly in the same habitats and with the same abiotic conditions.

*Our response: we agree with the reviewer that this is an important point that deserves more details. Accordingly, we have included a map of the study areas showing the location of carcasses (see new Fig. 1) and a table with the GPS coordinates of them (see new Table S1). Moreover, we have added some words to the main text to explain that we were very careful in selecting the carcass sites, so the microhabitat surrounding all of them was similar (“Carcasses were randomly placed in southern-oriented sites with intermediate vegetation cover [42]”; see lines 154-155 of the new version of the ms).

2. Experimental fox carrion was not eaten by vertebrate scavengers (except golden eagle). However authors noticed many incidents of visiting of experimental carrion by many vertebrate species. Could you describe this contact with more details? Has any physical contact with carrion occurred (jerking, moving etc.)?

*Our response: please note that we detected “some”, but not “many”, visits of few vertebrate scavenger species to carcass sites. In particular, cameras recorded 22 photographs of 5 vertebrate scavenger species in one study area (i.e., 2.2 photos/carcass and 1.3 species/carcass as average) and 20 photographs of 3 vertebrate scavenger species in the other study area (i.e., 2 photos/carcass and 0.4 species/carcass as average), as we already mentioned in results (lines 236-237 and 247-248 in the new version). According to the reviewer’s suggestion, we specify in the new version of the ms that “No other vertebrate species were recorded to contact physically with carcasses.” (Espuña study area; lines 245-246) and that “No vertebrate species were recorded to contact physically with carcasses, except a group of four wild boar piglets that probably touched one carcass. In this case, however, the potential contact was not sufficient to move the carcass.” (Bebedor study area; lines 252-255).

3. I have some hesitation related to quite low number of repetitions in the experiment. The information about detailed position of particular experimental carrions (suggested above) may tell us that material from each carrion is really comparable to others.

*Our response: we thank the reviewer for identifying which could be a potential limitation of our study. Getting fox carcasses for this kind of experiments is a difficult task, and that is probably why the scavenging patterns of fox carcasses have hardly been studied to date. Thus, as explained in our response to the first question above, we were very careful in selecting the sites in which we placed carcasses in order to make them comparable within and between study areas. In fact, our GLMM analyses confirmed that carcass identity was not an important variable to explain the variability in our response variables; in contrast, study area and, mostly, visit explained an important part of such variability.

4. I’m not sure (in contrary to authors…) that evisceration of carrion is without serious influence on the insect succession. Did authors do any tests for comparison of insect succession for eviscerated vs. eviscerated fox carrion? If not, it should be mentioned in the text.

*Our response: again, the reviewer highlights a good point. We always avoided exposing the open part of the carcasses to minimize the effect of the evisceration on scavenging patterns. We now specify this in lines 153-154. In any case, we feel that we already were cautious on this regard. As we discussed in the previous version, “It should also be noted that we used necropsied and eviscerated red fox carcasses, in order to minimise the risk of pathogen transmission [41]. The potential effect of the use of eviscerated carcasses on the patterns of carrion consumption by both vertebrate and invertebrate scavengers deserves further study. However, several studies have demonstrated that the rate of decomposition and the pattern of insect colonisation do not differ between intact and necropsied and/or wounded carcasses [75, 76]. In this sense, the composition of the insect fauna recorded during our study was overall similar to that reported from non-eviscerated, small- and medium-sized pig carcasses which were protected from vertebrate scavenging [22, 30]. Furthermore, our experimental carcasses were colonised by blow flies within the first three days after carcass exposure (Fig. 1). This is in line with the early colonisation patterns of carrion during winter, when the arrival of blow flies can sometimes be delayed a few days, in contrast to warmer months, when carrion can be colonised within hours after death [30, 63].” (lines 506-518 in the new version).

---

## [Decision Letter · Decision Letter 1]

19 Aug 2019

Avoidance of carnivore carcasses by vertebrate scavengers enables colonization by a diverse community of carrion insects

PONE-D-19-16398R1

Dear Dr. Martin-Vega,

We are pleased to inform you that your manuscript has been judged scientifically suitable for publication and will be formally accepted for publication once it complies with all outstanding technical requirements.

With kind regards,

Pierfilippo Cerretti, Ph.D.

Academic Editor

PLOS ONE

Additional Editor Comments (optional):

Dear Daniel,

The manuscript you submitted is now ready for publication, congratulations for this nice work!

Before resubmitting the final file, please make sure that each interval is divided by an en-dash, not by an hyphen (see for instance lines 317, 343, 346).

Best wishes,

Pierfilippo

Reviewers' comments:

Reviewer's Responses to Questions

**Comments to the Author**

1. If the authors have adequately addressed your comments raised in a previous round of review and you feel that this manuscript is now acceptable for publication, you may indicate that here to bypass the “Comments to the Author” section, enter your conflict of interest statement in the “Confidential to Editor” section, and submit your "Accept" recommendation.

Reviewer #2: All comments have been addressed

2. Is the manuscript technically sound, and do the data support the conclusions?

Reviewer #2: Yes

3. Has the statistical analysis been performed appropriately and rigorously? 

Reviewer #2: Yes

4. Have the authors made all data underlying the findings in their manuscript fully available?

Reviewer #2: Yes

5. Is the manuscript presented in an intelligible fashion and written in standard English?

Reviewer #2: Yes

6. Review Comments to the Author

Reviewer #2: New version of manuscript was corrected following previous comments. I have no more comments. Present version of manuscript is ready for publication.

7. PLOS authors have the option to publish the peer review history of their article (what does this mean?). If published, this will include your full peer review and any attached files.

Reviewer #2: No

---

## [Editor Report · Acceptance letter]

21 Aug 2019

PONE-D-19-16398R1 

Avoidance of carnivore carcasses by vertebrate scavengers enables colonization by a diverse community of carrion insects 

Dear Dr. Martín-Vega:

I am pleased to inform you that your manuscript has been deemed suitable for publication in PLOS ONE. Congratulations! Your manuscript is now with our production department. 

With kind regards,

on behalf of

Dr. Pierfilippo Cerretti 

Academic Editor

PLOS ONE